# AAPO: Enhancing the Reasoning Capabilities of LLMs with Advantage Momentum

## Abstract

Reinforcement learning (RL) has emerged as an effective approach for enhancing the reasoning capabilities of large language models (LLMs), especially in scenarios where supervised fine-tuning (SFT) falls short due to limited chain-of-thought (CoT) data. Among RL-based post-training methods, group relative advantage estimation, as exemplified by Group Relative Policy Optimization (GRPO), has attracted considerable attention for eliminating the dependency on the value model, thereby simplifying training compared to traditional approaches like Proximal Policy Optimization (PPO). However, we observe that existing group relative advantage estimation method still suffers from training inefficiencies, particularly when the estimated advantage approaches zero. To address this limitation, we propose Advantage-Augmented Policy Optimization (AAPO), a novel RL algorithm that optimizes the cross-entropy (CE) loss using advantages enhanced through a momentum-based estimation scheme. This approach effectively mitigates the inefficiencies associated with group relative advantage estimation. Experimental results on multiple mathematical reasoning benchmarks and model series demonstrate the superior performance of AAPO.

## 1 Introduction

Reinforcement learning (RL) has emerged as a powerful approach for enhancing the reasoning and decision-making capabilities of large language models (LLMs). While LLMs have demonstrated strong performance in both language understanding and generation tasks (Brown et al., 2020; Chowdhery et al., 2023; Touvron et al., 2023a; Zhao et al., 2023), traditional training strategies such as pre-training and supervised fine-tuning (SFT) (Radford et al., 2018; Bommasani et al., 2021; Liu et al., 2023) often fall short in enabling effective chain-of-thought (CoT) (Wei et al., 2022) reasoning for complex decision-making tasks. To address this limitation, recent research has explored RL-based training paradigms, which have shown considerable empirical success in specialized domains such as mathematical reasoning. Models including GPT-o1 (OpenAI, 2024), DeepSeek-R1 (Guo et al., 2025), and QwQ (Qwen Team, 2024) exemplify this promising direction, demonstrating the potential of RL to substantially improve the reasoning abilities of LLMs.

A recent advancement in LLM post-training with RL is the introduction of novel methods for advantage estimation, a technique for quantifying how favorable a specific action is in a given state. In this context, the term advantage measures the relative benefit of an action compared to the average in that state, thereby providing an informative learning signal during training. Traditionally, approaches such as Proximal Policy Optimization (PPO) (Schulman et al., 2017), exemplified by InstructGPT (Ouyang et al., 2022), rely on a value model to estimate the advantage. Although PPO offers stable and reliable performance, the need to maintain a separate value model leads to substantial consumption of GPU resources.

In contrast to conventional approaches, the group relative advantage estimation method was originally proposed in Group Relative Policy Optimization (GRPO) (Shao et al., 2024) and has since been widely adopted for enhancing reasoning capabilities in LLMs. Group relative advantage estimation method removes the need for value models entirely by evaluating responses relative to the average within a group of sampled responses. This approach significantly reduces GPU memory usage and computational costs while maintaining competitive performance in downstream reasoning tasks. Its effectiveness is further evidenced by the strong performance of modern reasoning mod-

els such as DeepSeek-R1 (Guo et al., 2025), which highlights the effectiveness of group relative advantage estimation in balancing computational efficiency and performance robustness. Several extensions have further refined this paradigm. Decoupled Clipping and Dynamic sAmpling Policy Optimization (DAPO) (Yu et al., 2025) improves training on long CoT sequences through token-level gradient estimation and relaxed clipping strategies. Dr. GRPO (Liu et al., 2025) introduces an unbiased optimization method that enhances token efficiency, while Group Policy Gradient (GPG) (Chu et al., 2025) further simplifies the learning process by removing surrogate losses and eliminating the reference model. While these methods vary in implementation specifics, they all share a common foundation in the principle of group relative advantage estimation for effective LLMs post-training. We provide a comprehensive discussion of GRPO and GPG in Section 3.

Although the above approaches have significantly advanced RL in the post-training stage and enhanced the reasoning capabilities of LLMs beyond what SFT can achieve, the practical limitations associated with the group relative advantage estimation method remain unresolved. As shown in the left panel of Figure 1, when the group relative advantage estimation method (Shao et al., 2024; Chu et al., 2025; Liu et al., 2025) is adopted, the advantage may approach when the rewards within a group exhibit low variance, resulting in zero gradient and, consequently, no parameter updates. Conversely, when rewards among responses within the group vary significantly, the resulting advantages can exhibit high variance, potentially leading to unstable or unintended gradient ascent. Both scenarios deviate from the desired optimization trajectories.

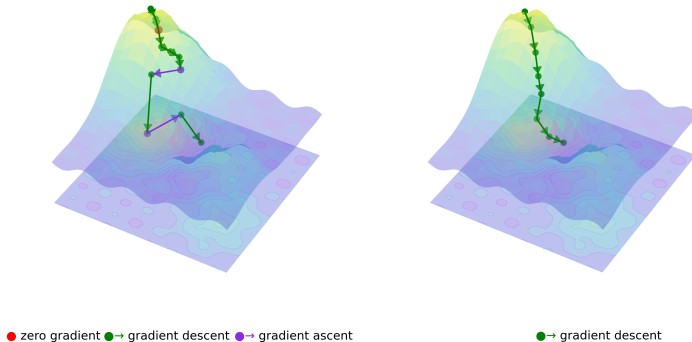

Figure 1: **Left**: An example convergence trajectories of policy optimization with group relative advantage estimation method proposed in GRPO. Red node denotes gradient equals to zero. Purple arrow denotes gradient ascent. Green arrow denotes gradient descent. **Right**: An example of ideal convergence trajectories of policy optimization.

In this work, to address the challenges of policy optimization from the group relative advantage estimation method mentioned above, we propose a novel RL algorithm Advantage-Augmented Policy Optimization (AAPO), which mitigates the issues demonstrated in Figure 1 by estimating the advantage with advantage momentum. Advantage momentum is defined as the distance between the rewards of responses from the policy model and those of the responses from the reference model. This approach incorporates a reference gradient into the original gradient, thus providing a reliable optimization signal that reflects the overall direction of improvement, even when the advantages approach zero. Experiments on several representative mathematical reasoning benchmarks demonstrate the effectiveness and robustness of AAPO.

In summary, our main contributions are as follows:

- We delve into the optimization behavior of current RL algorithms adopting the group relative advantage estimation method, in the context of post-training LLMs with RL, with a particular emphasis on potential issues related to advantage estimation during the optimization.

- We propose Advantage-Augmented Policy Optimization, which mitigates the issues of advantage estimation with advantage momentum by taking a comparison with the reference model.

- Experiments have demonstrated that AAPO achieves superior performance across different model series and mathematical reasoning benchmarks.

## 2 RELATED WORK

**Large Language Models** LLMs such as PaLM 2 (Anil et al., 2023) and Llama (Touvron et al., 2023a;b) achieved impressive performance in understanding and generation tasks, largely due to scaling laws, high-quality pre-train corpus, and optimized training techniques (Chen et al., 2024; Ouyang et al., 2022; Gunel et al., 2021; Wang et al., 2023). However, these models still struggle with complex reasoning. In contrast, recent LLMs such as GPT-o1 (OpenAI, 2024), DeepSeek-R1 (Guo et al., 2025), and QwQ (Qwen Team, 2024) show strong reasoning capabilities, particularly in mathematical tasks. These gains are primarily attributed to RL-based post-training, which significantly outperforms traditional SFT.

**Reinforcement Learning for LLMs** Early RL-based post-training methods (Christiano et al., 2017; Song et al., 2024; Ouyang et al., 2022; Ziegler et al., 2019) mainly relied on human-labeled preference data and reward models to evaluate responses. PPO (Schulman et al., 2017) estimates state values via a value model for advantage estimation while using a reward model to assign rewards. DPO (Rafailov et al., 2023) uses paired preference data to encourage preferred responses and suppress undesired ones. However, annotating preferences and training reward models are resource intensive. Moreover, the scarcity of explicit reasoning data limits the enhancement of models' reasoning capabilities via these RL methods. DeepSeek-R1 is the first to report the use of RL to enable extended CoT generation and the emergence of the so-called "aha moment" (Guo et al., 2025). Specifically, DeepSeek-R1 adopts GRPO (Shao et al., 2024), which estimates advantages through in-group comparisons, thereby enhancing the feasibility of aligning models for reasoning tasks. Yet, the limitations of existing advantage estimation remain underexplored. In this work, we introduce AAPO, which redefines advantage estimation by incorporating advantage momentum, effectively addressing the issues of the group relative advantage estimation method in policy optimization.

**Advantage estimation in Reinforcement Learning** Previous RL approaches (Schulman et al., 2017; 2015; Haarnoja et al., 2018) estimate the advantage using either Monte Carlo returns (Williams, 1992), Temporal-Difference (TD) (Sutton et al., 1999) errors, or Generalized Advantage Estimation (GAE) (Schulman et al., 2016). While these critic-based estimators form the backbone of algorithms such as PPO, maintaining a separate value network becomes computationally expensive when the policy is implemented as an LLM. Group relative advantage estimation method proposed by GRPO (Shao et al., 2024) computes the advantage by comparing each response's reward to the group mean, reducing the resources needed. However, it introduces new optimization challenges, as discussed in the introduction section, that have yet to be investigated. In this work, we propose a novel advantage estimation method that incorporates advantage momentum and optimizes the cross-entropy loss to better enhance the reasoning capabilities of LLMs.

## 3 PRELIMINARY

This section provides a preliminary overview of group-based relative advantage estimation methods for RL-based training by briefly introducing GRPO (Shao et al., 2024) and GPG (Chu et al., 2025). In the context of the GRPO algorithm, it circumvents the dependency on the value model which is commonly required in PPO (Schulman et al., 2017) by estimating advantage within grouped samples. It utilizes a rule-based reward system to score responses generated by LLMs. Furthermore, GRPO retains the clipping strategy from PPO to prevent excessively large policy updates and leverages the reference model $\pi_{ref}$ to compute the current Kullback-Leibler (KL) divergence, thereby ensuring the stability and integrity of the model during training. Specifically, for each question $q$, GRPO samples a group of responses $O = \{o_1, o_2, \cdots, o_G\}$ from the old policy $\pi_{\theta_{old}}$ and optimizes the policy model $\pi_\theta$ by maximizing the following objective:

$$
\mathcal{J}_{\mathrm{GRPO}}(\theta) = \mathbb{E}_{(q,a)\sim\mathcal{D},\{o_i\}_{i=1}^G\sim\pi_{\theta_{old}}(\cdot|q)} \frac{1}{G}\sum_{i=1}^G \frac{1}{|o_i|}\sum_{t=1}^{|o_i|}
$$

$$
\left\{ \min\left[ r_{i,t}(\theta)\hat{A}_{i,t}^{\mathrm{GRPO}}, \mathrm{clip}\left(r_{i,t}(\theta), 1-\varepsilon, 1+\varepsilon\right)\hat{A}_{i,t}^{\mathrm{GRPO}} \right] - \beta\mathbb{D}_{KL}\left[\pi_\theta||\pi_{ref}\right] \right\},
$$

(1)

where $\varepsilon$ and $\beta$ are hyper-parameters to control the clip boundary and the KL divergence penalty coefficient, respectively. In this context, $\hat{A}_{i,t}^{\mathrm{GRPO}}$ represents the advantage, derived exclusively from

the relative reward of the responses within the same group. $r_{i,t}(\theta)$ represents the likelihood ratio between the current policy $\pi_\theta$ and the old policy $\pi_{\theta_{\text{old}}}$. Typically, the likelihood ratio $r_{i,t}(\theta)$ and the advantage $\hat{A}_{i,t}^{\text{GRPO}}$ are calculated using the following formula:

$$r_{i,t}(\theta) = \frac{\pi_\theta\left(o_{i,t} \mid q, o_{i,<t}\right)}{\pi_{\theta_{\text{old}}}\left(o_{i,t} \mid q, o_{i,<t}\right)}, \quad \hat{A}_{i,t}^{\text{GRPO}} = \frac{r_i - \text{mean}(\{R_i\}_{i=1}^G)}{\text{std}(\{R_i\}_{i=1}^G)}. \tag{2}$$

Recent work GPG (Chu et al., 2025) proposes directly optimizing the original RL objective, yielding improved performance over GRPO. While the advantage estimation method used in GPG is similar to that of GRPO, it further emphasizes the advantage of responses that are valid to gradient estimation. Generally speaking, the core objective of GPG is to optimize the following objective:

$$\mathcal{J}_{\text{GPG}}(\theta) = \mathbb{E}_{(q,a)\sim\mathcal{D},\{o_i\}_{i=1}^G}\left[\frac{1}{\sum_{i=1}^G |o_i|}\sum_{i=1}^G\sum_{t=1}^{|o_i|}\left(-\log\pi_\theta\left(o_{i,t} \mid q, o_{i,<t}\right)\hat{A}_{i,t}^{\text{GPG}}\right)\right], \tag{3}$$

where $\hat{A}_{i,t}^{\text{GPG}} = \frac{r_i - \text{mean}(\{R_i\}_{i=1}^G)}{F_{norm}}$ and $F_{norm}$ could be 1 or $\text{std}(\{R_i\}_{i=1}^G)$. However, the rewards of the responses within a group may exhibit low variance. This suggests that $\hat{A}_{i,t}^{\text{GPG}}$ in GPG encounters the same challenges as $\hat{A}_{i,t}^{\text{GRPO}}$ in GRPO.

In this work, we propose a novel algorithm AAPO to address the issues caused by the group relative advantage estimation method in the policy optimization process. We also provide an in-depth analysis of both the prevailing advantage estimation method adopted in (Shao et al., 2024; Yu et al., 2025; Chu et al., 2025) and our proposed AAPO.

## 4 ADVANTAGE-AUGMENTED POLICY OPTIMIZATION

Inspired by the optimization behavior found in current RL algorithms (Shao et al., 2024; Liu et al., 2025; Yu et al., 2025; Chu et al., 2025), our objective is to mitigate the issues that arise in the policy optimization process, overcoming the challenge where advantage estimation tends to approach zero or bad advantage estimation in the later steps of RL training. Drawing further inspiration from the well-established Adam (Kingma & Ba, 2015) and the recent GPG algorithm (Chu et al., 2025), which optimizes the RL objective directly, thus avoiding the surrogate loss function, we propose a novel algorithm, Advantage-Augmented Policy Optimization (AAPO), which directly optimizes the cross-entropy (CE) loss enhanced by augmented advantage, which is driven by the advantage momentum. In contrast to previous approaches (Shao et al., 2024; Chu et al., 2025; Liu et al., 2025), AAPO leverages advantage amplification by performing group-based sampling for both the policy model $\pi_\theta$ and the reference model $\pi_{ref}$. Specifically, we calculate the reward for each response generated by the policy model $\pi_\theta$, evaluate the relative advantage of each response within its group $G$, and compare these rewards $r_{\theta_i}$ with those $r_{ref_i}$ obtained from the reference model $\pi_{ref}$. This method improves the effectiveness of the RL training step by preventing the advantage from approaching zero and exhibiting high variance. Formally, AAPO optimizes the policy model $\pi_\theta$ by minimizing the following objective:

$$\mathcal{J}_{\text{AAPO}}(\theta) = \mathbb{E}_{(q,a)\sim\mathcal{D},\{o_i\}_{i=1}^G}\left[\frac{1}{\sum_{i=1}^G |o_i|}\sum_{i=1}^G\sum_{t=1}^{|o_i|}\left(-\log\pi_\theta\left(o_{i,t} \mid q, o_{i,<t}\right)\hat{A}_{i,t}^*\right)\right], \tag{4}$$

where

$$\hat{A}_{i,t}^* = \frac{r_{\theta_i} - \text{mean}(r_\theta)}{\text{std}(r_\theta)} + \text{clip}(\underbrace{r_{\theta_i} - r_{ref_i}}_{\textbf{Advantage momentum}}, \delta_{\text{low}}, \delta_{\text{high}}). \tag{5}$$

Based on equation 4, it is straightforward to derive its gradient:

$$\nabla_\theta\mathcal{J}_{\text{AAPO}}(\theta) = -\mathbb{E}_{(q,a)\sim\mathcal{D},\{o_i\}_{i=1}^G}\left[\frac{1}{\sum_{i=1}^G |o_i|}\sum_{i=1}^G\sum_{t=1}^{|o_i|}\hat{A}_{i,t}^* \cdot \nabla_\theta\log\pi_\theta\left(o_{i,t} \mid q, o_{i,<t}\right)\right], \tag{6}$$

where the augmented advantage $\hat{A}_{i,t}^*$ functions as a constant coefficient that scales the gradient.

**Definition** *For a group $\mathcal{G}$ containing sampled responses $O = \{o_1, o_2, \cdots, o_{\mathcal{G}}\}$, the empirical AAPO loss is defined as $\mathcal{L}_{\mathcal{G}}(\theta) = \frac{1}{N_{\mathcal{G}}} \sum_{o \in \mathcal{G}} [-\log \pi_\theta(o)\hat{A}^*]$, where $\pi_\theta$ is the policy model, $N_{\mathcal{G}} = \sum_{o \in \mathcal{G}} |o|$ is the total number of tokens in the group. We further define the expected objective as the expectation of the empirical loss over all possible groups $\mathcal{L}(\theta) = \mathbb{E}_{\mathcal{G} \sim \pi_\theta}[\mathcal{L}_{\mathcal{G}}(\theta)]$.*

**Theorem 1.** *(Stability) Since the rewards are bounded, the group standard deviation satisfies $0 \leq \sigma_{min} \leq \sigma$, and the log-likelihood gradients are bounded as $||\nabla_\theta \log \pi_\theta(o)|| \leq M$. Then, Each gradient step with learning rate $\eta_k$ satisfies $||\theta_{k+1} - \theta_k|| \leq \eta_k M B$, where $B = \frac{R_{max} - R_{min}}{\sigma_{min}} + \max(|\delta_{\text{low}}|, |\delta_{\text{high}}|)$ is the uniform bound on the AAPO weights. The expected objective is bounded from $\mathcal{L}(\theta) \geq -B \log |\mathcal{V}|$, where $|\mathcal{V}|$ is the vocabulary size. Hence, AAPO training is stable: the objective cannot diverge to $-\infty$ and parameter updates are always finite. Proof in Appendix B.*

**Theorem 2.** *(Convergence) Assume that the stochastic gradient is unbiased and that the per-sample gradient has bounded second moment. Let the step sizes satisfy the Robbins–Monro conditions $\eta_k > 0$, $\sum_k \eta_k = \infty$, $\sum_k \eta_k^2 < \infty$. AAPO converges to a stationary point of its expected objective $\liminf_{k \to \infty} \mathbb{E}\left[||\nabla \mathcal{L}(\theta_k)||^2\right] = 0$. Moreover, if a constant step size $\eta < \frac{1}{BL_0}$ is used, where $L_0$ is the smoothness constant of $-\log \pi_\theta(o)$, then the iterates converge to a neighborhood of stationarity $\limsup_{K \to \infty} \frac{1}{K} \sum_{k=1}^K \mathbb{E}\left[||\nabla \mathcal{L}(\theta_k)||^2\right] \lesssim \mathcal{O}(\eta) + \mathcal{O}\left(\frac{1}{N_{\mathcal{G}}}\right)$. Proof in Appendix B.*

From the formulation of the augmented advantage $\hat{A}_{i,t}^*$, it is evident that as the policy is optimized, the reward $r_{\theta_i}$ of the responses sampled from the policy $\pi_\theta$ within a group increases. Consequently, the distance between these rewards $r_{\theta_i}$ and those $r_{ref_i}$ of the responses sampled from the reference model $\pi_{ref}$ will also widen, since the parameters of the reference model remain frozen throughout the training process of AAPO. In later steps of RL, the responses sampled from the policy tend to be of high quality, causing the relative advantages $\hat{A}_{i,t}^{\text{GRPO}}$ within the group to approach zero. If training continues using the original advantage estimation method in equation 2, the resulting gradients will approach zero, leading to significantly reduced training efficiency. However, under the proposed method augmenting advantage with advantage momentum in equation 5, even when the group relative advantage approaches zero, the rewards of the policy samples remain higher than those of the reference samples. This ensures a nonzero advantage $\hat{A}_{i,t}^*$ in AAPO as discussed in Section 5.2, thereby maintaining informative gradients for continued policy optimization.

# 5 ANALYSIS OF AAPO

## 5.1 DEEP ANALYSIS OF GROUP RELATIVE ADVANTAGE ESTIMATION

As shown in equation 2, current advantage estimation methods in recent RL approaches (Shao et al., 2024; Yu et al., 2025; Liu et al., 2025) that eliminate the dependency on a value model predominantly adopt this form of computation. We now present a rigorous analysis into the underlying phenomena induced by this advantage estimation method: (1) Phenomenon 1: What are the implications when all responses within a group are similarly good (or all of high quality)? (2) Phenomenon 2: What are the implications when all responses within a group are similarly bad (or all of low quality)?

To rigorously address the aforementioned questions, we proceed with a systematic, step-by-step analysis. For the sake of mathematical convenience in the proof, we limit our discussion to cases where all responses are similarly good in Phenomenon 1 and similarly bad in Phenomenon 2, respectively, as the proof for all of high quality and low quality responses follows the same structure.

**Phenomenon 1** Considering that all responses are similarly good, which implies that the reward for each response is nearly the same, this indicates $\forall i, j \in \{1, 2, 3, \cdots, G\} \wedge i \neq j$, $r_i \approx r_j$, their respective advantage, as estimated according to equation 2, can be expressed in the following form:

$$\hat{A}_{i,t}^{\text{GRPO}} = \frac{r_i - \text{mean}(\{R_i\}_{i=1}^G)}{\text{std}(\{R_i\}_{i=1}^G)} \to 0. \tag{7}$$

As expressed in equation 7, the advantage of each response approaches zero in this case. For illustrative purposes, we use the loss function of GRPO (Yu et al., 2025) as an example. As mentioned in

DAPO (Yu et al., 2025), removing the KL divergence in GRPO could further improve the optimization. The KL divergence, summation and averaging operators in equation 1 are omitted to facilitate clarity in this context. Under these conditions, the gradient formula of GRPO is formally given by:

$$\nabla_\theta \mathcal{J}_{\text{GRPO}}(\theta) = A_{i,t}^{\text{GRPO}} \nabla_\theta \log \pi_\theta(o_{i,t} \mid q, o_{i,<t}). \qquad (8)$$

As illustrated in the computation of equation 8, when the advantage of each response approaches zero, the corresponding gradient also decreases to zero. This implies that the gradient update for the policy becomes negligible for this training step, resulting in very low training efficiency. Similar issues are observed in methods such as GRPO, DAPO, GPG (Chu et al., 2025), and Dr. GRPO (Liu et al., 2025). Nonetheless, the observation that all responses are associated with similarly high reward does not unequivocally imply that the policy has been sufficiently optimized; it may alternatively reflect a high variance (Roelofs et al., 2019; Yu et al., 2022) in the current policy, suggesting that the policy has converged to a sub-optimum, performing well only on a narrow category of questions. However, this phenomenon is frequently observed during the later steps of RL training if the advantage estimation method in GRPO is adopted.

**Phenomenon 2**   Similar to the scenario in Phenomenon 1 where all responses are similarly bad, when all responses are similarly bad, the reward assigned to each response tends to be similar. As a result, the relative advantage of each response approaches zero, leading to a zero gradient during policy updates, which is computed using equation 8. Consequently, the efficiency of this training step becomes significantly low. Such a phenomenon is frequently observed when the policy is confronted with input samples that exhibit inherently high complexity or ambiguous representation.

**Analysis Conclusion**   Based on our in-depth analysis of the two phenomena where generated responses are similarly good and similarly bad within a group, we observed that the gradient tends to approach zero. This results in extremely low training efficiency during RL training.

**Above phenomena can be generalized**   When the rewards of responses within any given group are identical or highly similar, regardless of whether the responses are all good or all bad, the gradient approaches zero, rendering the gradient update nearly ineffective in training. This issue becomes particularly pronounced in the later steps of RL training, where generated responses are consistently of high quality, and the corresponding advantage approaches zero. This indicates that the training efficiency progressively declines as RL training progresses. Motivated by this insight, we propose an advantage-augmented RL algorithm AAPO to address the aforementioned phenomena.

## 5.2 Understanding the effectiveness of AAPO

As discussed in Section 5.1, the previous advantage estimation method (as expressed in equation 2) in RL algorithms (Shao et al., 2024; Chu et al., 2025; Liu et al., 2025) can lead to the advantage value approaching zero, which in turn causes the magnitude of gradient updates to diminish accordingly. In this section, we provide a comprehensive analysis of why our proposed advantage-augmented method can effectively mitigate these issues.

**Analysis 1**   Consider a general situation, which naturally encompasses the two phenomena in Section 5.1. In the later steps of RL training, once the policy has already acquired relatively easier-to-learn features, it often struggles to learn more complex ones. During this phase, when the policy $\pi_\theta$ samples a group of responses, the reward associated with each sample tends to be similar. Consequently, the relative advantage computed according to equation 2 approaches zero. To address this issue, we propose AAPO, which estimates the advantage $\hat{A}_{i,t}^*$ with advantage momentum following equation 5. Since the capability of the reference model $\pi_{ref}$ remains unchanged while the policy model $\pi_\theta$ improves progressively throughout AAPO training, the quality of responses generated by the policy model $\pi_\theta$ exceeds that of the reference model $\pi_{ref}$ over time. By measuring the distance between the rewards of two groups of responses $O_\theta = \{o_{\theta_1}, o_{\theta_2}, \cdots, o_{\theta_G}\}$ and $O_{ref} = \{o_{ref_1}, o_{ref_2}, \cdots, o_{ref_G}\}$ from $\pi_{ref}$ generated by the policy model $\pi_\theta$ and the reference model $\pi_{ref}$, respectively, we can calculate the augmented advantages of each response in $O_\theta$ via equation 5. This prevents the advantages from approaching zero and ensures that the gradients used to update the policy remain informative and effective. This analysis can be generalized to any situation where the rewards of the responses in the group are similar, whether good or not.

**Analysis 2**   Group relative advantage estimation (equation 2) can be problematic in the presence of two types of asymmetric response distributions. In one case, where most responses from the policy

$\pi_\theta$ are high quality except for a single outlier, the relatively worse response may receive a negative advantage and contribute an opposing gradient, increasing variance. Although this response may still be correct under multi-dimensional reward rules, such as both format and correctness, it is penalized due to its format. This misalignment between reward attribution and the true value of a response increases the risk of reward hacking(Everitt et al., 2021; Pan et al., 2022). In the opposite case, where most responses are low quality and one is relatively better, the better one receives an excessively high advantage despite possibly low absolute quality, leading to biased updates and suboptimal convergence. AAPO augments advantage estimation with advantage momentum, addresses both issues: it boosts underappreciated yet valuable responses and suppresses misleading ones that exhibit a high estimated advantage. As a result, AAPO reduces variance and the risk of reward hacking, and improves the optimization stability of the policy.

**Analysis 3**     When policy optimization with AAPO reaches a global optimum, which means $\forall i, j \in \{1, 2, 3, \cdots, G\} \wedge i \neq j, \ r_i \approx r_j$, the objective in equation 6, which consists of the CE loss and the augmented advantage, exhibits a specific and predictable behavior as detailed in our derivation below (for clarity and notational convenience, we omit the expectation operator). The optimization exhibits slight oscillations near the optimum, avoiding large gradient updates.

$$\mathcal{J}_{\text{AAPO}}(\theta) \approx \frac{1}{\sum_{i=1}^{G} |o_i|} \sum_{i=1}^{G} \sum_{t=1}^{|o_i|} \left( -\log \pi_\theta \left( o_{i,t} \mid q, o_{i,<t} \right) \cdot \text{clip}(r_{\theta_i} - r_{ref_i}, \delta_{\text{low}}, \delta_{\text{high}}) \right)$$

$$\leq \delta_{\text{high}} \cdot \frac{1}{\sum_{i=1}^{G} |o_i|} \sum_{i=1}^{G} \sum_{t=1}^{|o_i|} \left( -\log \pi_\theta \left( o_{i,t} \mid q, o_{i,<t} \right) \right)$$

## 6    EXPERIMENTS

### 6.1    EXPERIMENTAL SETUP

Most recent RL algorithms (Shao et al., 2024; Yu et al., 2025; Hu, 2025; Liu et al., 2025) struggle to make LLMs perform better at solving mathematical problems, which requires the model to think in the CoT (Wei et al., 2022) format before deciding the final answer. We choose open-rs (Dang & Ngo, 2025) as our training dataset for DeepSeek-R1-Distill-Qwen-1.5B base model, because the data in this dataset cover various types and difficulty levels of mathematical problems that are highly representative. To further provide a fair and rigorous evaluation of the effectiveness of our proposed AAPO, Qwen2.5-Math-7B model is chosen as the base model and subsequently trained on more challenging simplelr_qwen_level3to5 dataset (Zeng et al., 2025). In addition to the different model sizes, we also validate the effectiveness of AAPO on Llama series models.

In our experiment setting, we set the clip parameters $\delta_{\text{low}}$ and $\delta_{\text{high}}$ to be -0.2 and 0.28, respectively. We train all base models under the AAPO following the training process depicted in Algorithm 1. All rule-based reward functions adopted in our experiments are simple and straightforward. More training details about rule-based reward functions and the system prompt are provided in Appendix C. To evaluate the extent to which our proposed AAPO algorithm can enhance the reasoning capabilities of the model, we select AIME24, MATH-500 (Hendrycks et al., 2021; Lightman et al., 2024), AMC23, Minerva (Lewkowycz et al., 2022), and OlympiadBench (Huang et al., 2024) as evaluation benchmarks. Our evaluation framework utilizes a well-established and community-vetted codebase, maintaining consistency with widely adopted implementations.

### 6.2    RESULTS

As shown in Table 1, taking model DeepSeek-R1-Distill-Qwen-1.5B as our base model, the application of our proposed AAPO enables the base model to achieve the SOTA performance on Minerva, the second-best performance on MATH-500, AMC23 and OlympiadBench. When averaging scores across all benchmarks, the resulting AAPO-1.5B model achieves an overall SOTA performance. By direct comparison under the same training data, AAPO-1.5B achieves improvements of **2.7%** and **2.0%** over GRPO-1.5B and GPG-1.5B, respectively. It is worth noting that our AAPO-1.5B achieves performance superior to Still-3-1.5B-Preview (Team, 2025), despite the fact that Still-3-1.5B-Preview benefits from a larger training dataset that consists of long CoT reasoning data distilled from the DeepSeek-R1 (Guo et al., 2025) model and performs RL training with

specified reward strategies. After employing our proposed AAPO on the Qwen2.5-Math-7B model, AAPO-7B achieves the overall SOTA performance compared to other methods. AAPO-7B achieves improvements of **12.2%**, **10.3%** over SimpleRL-Zero-7B (Zeng et al., 2025), GPG-7B (Chu et al., 2025), respectively, under identical training data. Furthermore, AAPO-7B also outperforms other models such as Eurus-2-7B-PRIME (Cui et al., 2025), OpenReasoner-Zero-7B (Hu et al., 2025), despite the fact that these baselines are trained with more high-quality data or data distilled from the DeepSeek-R1 model. This suggests that the effectiveness can be mainly attributed to the design of our proposed AAPO rather than to the scale or quality of the training data. AAPO also demonstrates superior performance compared to GRPO and GPG on Llama series models under the same training and evaluation settings as reported in Table 1. Compared to GRPO and GPG, AAPO achieves absolute improvements of **3.5%**, **2.5%**, and **4.6%**, **2.6%** on Llama 1B and 3B models, respectively. It is worth noting that evaluation results may be influenced by the computational device type. We have also provided the original results of each model from the corresponding papers in Appendix E.

| Model | AIME24 | MATH-500 | AMC23 | Minerva | OlympiadBench | Avg. |
|---|---|---|---|---|---|---|
| *Llama 1B Models* | | | | | | |
| Llama-3.2-1B-Instruct[†] | 0.0 | 11.8 | 2.5 | 1.8 | 3.7 | 4.0 |
| +GRPO[†] (Shao et al., 2024) | 0.0 | 19.4 | 12.5 | 3.7 | 4.3 | 8.0 |
| +GPG[†] (Chu et al., 2025) | 0.0 | 21.2 | 17.5 | 1.8 | 4.7 | 9.0 |
| +AAPO (Ours) | 10.0 | 25.0 | 12.5 | 4.0 | 6.1 | **11.5** |
| *Llama 3B Models* | | | | | | |
| Llama-3.2-3B-Instruct[†] | 3.3 | 28.6 | 7.5 | 3.7 | 7.6 | 10.1 |
| +GRPO[†] (Shao et al., 2024) | 0.0 | 32.8 | 22.5 | 8.1 | 7.7 | 14.2 |
| +GPG[†] (Chu et al., 2025) | 6.7 | 40.0 | 15.0 | 8.8 | 11.6 | 16.2 |
| +AAPO (Ours) | 6.7 | 43.8 | 22.5 | 9.6 | 11.4 | **18.8** |
| *Qwen 1.5B Models* | | | | | | |
| DeepSeek-R1-Distill-Qwen-1.5B[†] | 33.3 | 84.4 | 70.0 | 30.9 | 50.8 | 53.9 |
| GRPO-1.5B[†] (Dang & Ngo, 2025) | 26.7 | 86.2 | 82.5 | 27.6 | 52.6 | 55.2 |
| GPG-1.5B[†] (Chu et al., 2025) | 36.7 | 83.4 | 75.0 | 29.8 | 53.2 | 55.6 |
| Still-3-1.5B-Preview[†] (Chen et al., 2025) | 40.0 | 85.5 | 72.5 | 30.5 | 53.9 | 56.5 |
| AAPO-1.5B (Ours) | 33.3 | 86.0 | 80.0 | 30.9 | 53.3 | **56.7** |
| *Qwen 7B Models* | | | | | | |
| Qwen2.5-Math-7B[†] (Yang et al., 2024) | 6.7 | 56.2 | 47.5 | 14.0 | 23.4 | 39.6 |
| SimpleRL-Zero-7B[†] (Zeng et al., 2025) | 30.0 | 77.4 | 57.5 | 30.5 | 38.1 | 46.7 |
| GPG-7B[†] (Chu et al., 2025) | 23.3 | 80.2 | 55.0 | 36.0 | 42.8 | 47.5 |
| OpenReasoner-Zero-7B[†] (Hu et al., 2025) | 20.0 | 80.8 | 65.0 | 29.4 | 46.2 | 48.3 |
| Eurus-2-7B-PRIME[†] (Cui et al., 2025) | 16.7 | 81.8 | 65.0 | 37.5 | 44.6 | 49.1 |
| Oat-Zero-7B[†] (Liu et al., 2025) | 30.0 | 81.2 | 65.0 | 34.9 | 43.4 | 50.3 |
| AAPO-7B (Ours) | 30.0 | 82.4 | 70.0 | 35.3 | 44.3 | **52.4** |

Table 1: Zero-shot pass@1 performance on mathematical reasoning benchmarks. [†] represents reproduced results with our best effort under the same settings. **Bold** and Underline indicate the best and the second-best performance in the corresponding category, respectively.

## 6.3 ABLATION STUDY

**Clip operation** To investigate the contribution of the clip operation to AAPO, we conduct additional ablation studies by removing the clip operation on both the 1.5B and 7B models. The results presented in Table 2 indicate that the performance of the optimized model without clip is inferior to that with the clip, suggesting that incorporating the clip operation effectively contributes to further improvements in optimization results. As illustrated in the right figure, the optimization process becomes more stable with the incorporation of the clip operation compared to the optimization process without it. The resulting AAPO-7B exhibits better performance on the benchmarks when the clip operation is adopted.

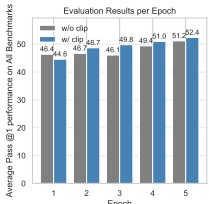

| Model | AIME24 | MATH-500 | AMC23 | Minerva | OlympiadBench | Avg. |
|---|---|---|---|---|---|---|
| AAPO-1.5B | 33.3 | 86.0 | 80.0 | 30.9 | 53.3 | 56.7 |
| AAPO-1.5B *w/o* clip | 33.3 | 85.0 | 82.5 | 29.0 | 53.3 | 56.6 |
| AAPO-7B | 30.0 | 82.4 | 70.0 | 35.3 | 44.3 | 52.4 |
| AAPO-7B *w/o* clip | 30.0 | 79.6 | 70.0 | 34.9 | 42.5 | 51.2 |

Table 2: Ablation study results. *w/o* indicates without clip operation on the advantage momentum. Zero-shot pass@1 performance on different mathematical benchmarks.

**Training process analysis**     As illustrated by the training curves in Figure 2, AAPO achieves optimization performance comparable to that of GPG, while exhibiting further improvements during the later steps of RL training. In the Format Reward and Reward figures, AAPO consistently attains higher reward than GPG in the later steps of the training. In addition, three Reward figures demonstrate significantly reduced fluctuations during the training process. AAPO also outperforms GRPO. Moreover, in the Response Length figure, our proposed AAPO demonstrates superior optimization, indicating higher training efficiency. The final results presented in Table 1 demonstrate that our proposed AAPO achieves superior performance across five mathematical reasoning benchmarks. To straightly and effectively analyze the training stability of AAPO, we calculated the variance of the training loss for AAPO as well as methods GRPO and GPG. $\text{Var}(\mathcal{L}_{\text{AAPO}}) = 3.6 \times 10^{-4}$, $\text{Var}(\mathcal{L}_{\text{GPG}}) = 3.9 \times 10^{-4}$ and $\text{Var}(\mathcal{L}_{\text{GRPO}}) = 3.53 \times 10^{-3}$. The results show that AAPO exhibits the lowest variance, indicating relatively stable training. Additionally, the results of the ablation experiments on clip operation demonstrate that AAPO can steadily improve model performance as training epochs progress. More experiment analysis can be found in Appendix D.

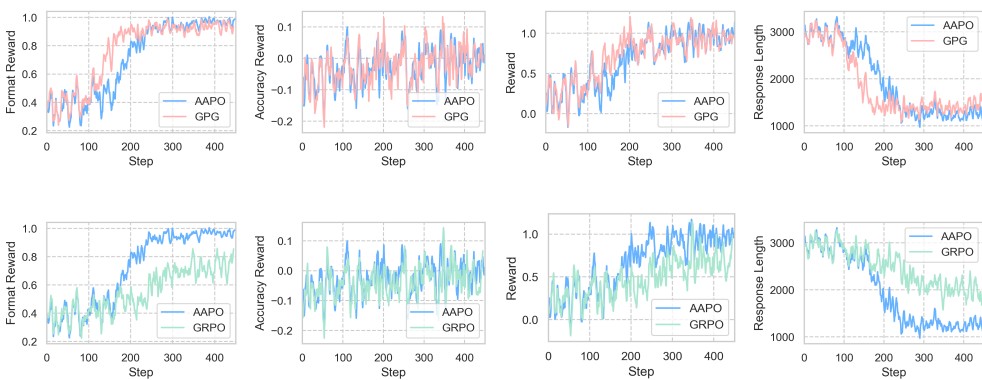

Figure 2: Training process of DeepSeek-R1-Distill-Qwen-1.5B on open-rs (Dang & Ngo, 2025) utilizing our proposed AAPO algorithm. Compared to GPG (Chu et al., 2025) and GRPO (Shao et al., 2024), AAPO demonstrates better stability during training and achieves superior performance in the final results as shown in Table 2.

## 7    CONCLUSION

In this paper, we conduct an in-depth analysis of the limitations inherent in the group relative advantage estimation method used by mainstream RL algorithms, such as GRPO, which would lead to optimization issues such as zero gradient and gradient ascent. To address these issues, we propose a novel RL algorithm Advantage-Augmented Policy Optimization (AAPO). By augmenting the group relative advantage estimation method with advantage momentum, our method effectively improves policy optimization performance in experimental benchmarks. Experimental results across several mathematical reasoning benchmarks and model series demonstrate that AAPO achieves the overall superior performance across several mathematical reasoning benchmarks.

**Reproducibility statement** Experimental details about training and evaluation are depicted in both Section 6 and Appendix C, our training code is implemented based on Transformers Reinforcement Learning framework (von Werra et al., 2020). All codes are available at `https://anonymous.4open.science/r/AAPO-review-01B9`.

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

# A ALGORITHM

**AAPO Training** The training process of AAPO can be summarized as follows: 1) Sample two groups of responses, $O_\theta$ and $O_{ref}$, of equal size from both the policy model $\pi_\theta$ and the reference model $\pi_{ref}$, respectively. 2) For each response in $O_\theta$, compute its group relative advantage and the advantage momentum with the corresponding response in $O_{ref}$. The values, calculated by equation 5, are then used to perform gradient updates according to equation 6. It is important to note that during the training process, the parameters of the reference model $\pi_{ref}$ remain frozen and do not undergo gradient updates. The training procedure is described in Algorithm 1.

---

**Algorithm 1** Advantage-Augmented Policy Optimization

---

1: **Input:** policy model $\pi_\theta$, reference model $\pi_{ref}$, group size $G$, reward functions $F = \{\text{format}, \text{accuracy}, \cdots\}$, reward functions' corresponding weights $W = \{w_{\text{format}}, w_{\text{accuracy}}, \cdots\}$, data batch $\mathcal{B}$, global step $\mathcal{S}_{gloabl}$
2: **for** $i = 1$ **to** $\mathcal{S}_{gloabl}$ **do**
3:     **for** $j = 1$ **to** $\mathcal{B}$ **do**
4:         1) sample $O_\theta = \{o_{\theta_1}, o_{\theta_2}, \cdots, o_{\theta_G}\}$ from $\pi_\theta$
5:         2) sample $O_{ref} = \{o_{ref_1}, o_{ref_2}, \cdots, o_{ref_G}\}$ from $\pi_{ref}$
6:         3) compute the rewards $R_\theta = \{R_\theta^{\text{format}}, R_\theta^{\text{accuracy}}, \cdots, R_\theta^{\cdots}\}$ and $R_{ref} = \{R_{ref}^{\text{format}}, R_{ref}^{\text{accuracy}}, \cdots, R_{ref}^{\cdots}\}$ for each response in $O_\theta$ and $O_{ref}$ using the reward functions in $F$, e.g., $R_\theta^{\text{format}} = \{r_{\theta_1}^{\text{format}}, r_{\theta_2}^{\text{format}}, \cdots, r_{\theta_G}^{\text{format}}\}^\top$ and $R_{ref}^{\text{format}} = \{r_{ref_1}^{\text{format}}, r_{ref_2}^{\text{format}}, \cdots, r_{ref_G}^{\text{format}}\}^\top$
7:         4) compute the weighted rewards $R_\theta^{\text{weighted}} = R_\theta W^\top$ and $R_{ref}^{\text{weighted}} = R_{ref} W^\top$
8:         5) compute the augmented advantage $\hat{A}_{i,t}^*$ following equation 5 for each sample in $O_\theta$
9:         6) compute loss for $O_\theta$ following equation 4
10:     **end for**
11:     7) update $\pi_\theta$ following equation 6
12: **end for**

---

# B PROOF

**Theorem 1.** *(Stability)* *Since the rewards are bounded, the group standard deviation satisfies $0 \leq \sigma_{min} \leq \sigma$, and the log-likelihood gradients are bounded as $||\nabla_\theta \log \pi_\theta(o)|| \leq M$. Then, Each gradient step with learning rate $\eta_k$ satisfies $||\theta_{k+1} - \theta_k|| \leq \eta_k MB$, where $B = \frac{R_{max} - R_{min}}{\sigma_{min}} + \max(|\delta_{\text{low}}|, |\delta_{\text{high}}|)$ is the uniform bound on the AAPO weights. The expected objective is bounded from $\mathcal{L}(\theta) \geq -B \log |\mathcal{V}|$, where $|\mathcal{V}|$ is the vocabulary size. Hence, AAPO training is stable: the objective cannot diverge to $-\infty$ and parameter updates are always finite.*

*Proof.* We restate the definitions used: (1) For any response $o$, the advantage $\hat{A}^*$ satisfies $|\hat{A}^*| \leq B = \frac{R_{max} - R_{min}}{\sigma_{min}} + \max(|\delta_{\text{low}}|, |\delta_{\text{high}}|)$, where $\sigma_{min} > 0$ is the minimal standard deviation ensured by group sampling. (2) The gradient of the log-policy is bounded: $||\nabla_\theta \log \pi_\theta(o)|| \leq M$, and $-\log \pi_\theta(o)$ is $L_0$-Lipschitz. (3) The vocabulary size satisfies $|\mathcal{V}| < \infty$. For a group $\mathcal{G}$, the gradient of the empirical loss is:

$$\nabla_\theta \mathcal{L}_\mathcal{G}(\theta) = \frac{1}{N_\mathcal{G}} \sum_{o \in O} \nabla_\theta [-\log \pi_\theta(o)] \hat{A}^*.$$

By assumption (1) and (2), we have

$$||\nabla_\theta \mathcal{L}_\mathcal{G}(\theta)|| \leq \frac{1}{N_\mathcal{G}} \sum_{o \in O} M \cdot B = MB.$$

Thus, one gradient update with learning rate $\eta_k$ yields

$$||\theta_{k+1} - \theta_k|| = \eta_k ||\nabla_\theta \mathcal{L}_\mathcal{G}(\theta)|| \leq \eta_k MB.$$

For any response $o$,

$$\mathbb{E}_{a \sim \pi_\theta} \left[ -\log \pi_\theta(o) \right] = H(\pi_\theta(\cdot \mid o_{<t})) \leq \log |\mathcal{V}|.$$

where $H(\cdot) = -\sum_x p(x) \log p(x)$ denotes Shannon entropy (Shannon, 1948). Multiplying by the bounded advantage $|\hat{A}^*| \leq B$ and averaging over tokens in the batch, we obtain

$$\mathcal{L}(\theta) \geq -B \log |\mathcal{V}|.$$

we conclude that parameter updates are bounded and the objective is lower bounded (which means the objective cannot diverge to $-\infty$). Therefore, AAPO training is stable and cannot diverge. $\square$

**Theorem 2.** *(Convergence)    Assume that the stochastic gradient is unbiased and that the per-sample gradient has bounded second moment. Let the step sizes satisfy the Robbins–Monro conditions $\eta_k > 0$, $\sum_k \eta_k = \infty$, $\sum_k \eta_k^2 < \infty$. AAPO converges to a stationary point of its expected objective $\liminf_{k \to \infty} \mathbb{E}\left[\|\nabla \mathcal{L}(\theta_k)\|^2\right] = 0$. Moreover, if a constant step size $\eta < \frac{1}{BL_0}$ is used, where $L_0$ is the smoothness constant of $-\log \pi_\theta(o)$, then the iterates converge to a neighborhood of stationarity $\limsup_{K \to \infty} \frac{1}{K} \sum_{k=1}^{K} \mathbb{E}\left[\|\nabla \mathcal{L}(\theta_k)\|^2\right] \lesssim \mathcal{O}(\eta) + \mathcal{O}\left(\frac{1}{N_\mathcal{G}}\right).$*

*Proof.* We adopt the same definitions as in the Proof of Theorem 1. Since $-\log \pi_\theta(o \mid q)$ is $L_0$-smooth (i.e., its gradient is $L_0$-Lipschitz) and the advantage is bounded by $B$, the empirical loss $\mathcal{L}_\mathcal{G}(\theta)$ is $L = BL_0$-smooth:

$$\|\nabla \mathcal{L}_\mathcal{G}(\theta) - \nabla \mathcal{L}_\mathcal{G}(\theta')\|_2 \leq L \|\theta - \theta'\|_2.$$

By the descent lemma for $L$-smooth functions, for any step size $\eta_k$,

$$\mathcal{L}_\mathcal{G}(\theta - \eta_k g) \leq \mathcal{L}_\mathcal{G}(\theta) - \eta_k \langle g, \nabla \mathcal{L}_\mathcal{G}(\theta) \rangle + \tfrac{L}{2} \eta_k^2 \|g\|^2.$$

Choosing $g = \nabla \mathcal{L}_\mathcal{G}(\theta_k)$ and requiring $\eta_k \leq 1/L$, we obtain

$$\mathcal{L}_\mathcal{G}(\theta_{k+1}) \leq \mathcal{L}_\mathcal{G}(\theta_k) - \tfrac{\eta_k}{2} \|\nabla \mathcal{L}_\mathcal{G}(\theta_k)\|^2.$$

Assume that, for all $\theta$, the per-sample loss $\ell(q, o; \theta) = -\log \pi_\theta(o \mid q) \hat{A}^*$ has a gradient with bounded second moment:

$$\sup_\theta \mathbb{E}_{(q,o) \sim \pi_\theta} \left[ \|\nabla_\theta \ell(q, o; \theta)\|^2 \right] \leq \sigma^2,$$

where $\sigma^2$ is a constant. Since $\nabla \mathcal{L}_\mathcal{G}(\theta_k)$ is the average of $N_\mathcal{G}$ i.i.d. samples conditional on $\theta_k$, we have

$$\mathbb{E}[\nabla \mathcal{L}_\mathcal{G}(\theta_k)] = \nabla \mathcal{L}(\theta_k), \quad \mathbb{E}\left[ \|\nabla \mathcal{L}_\mathcal{G}(\theta_k) - \nabla \mathcal{L}(\theta_k)\|^2 \right] \leq \frac{\sigma^2}{N_\mathcal{G}}.$$

Consequently (by variance decomposition),

$$\mathbb{E}\left[ \|\nabla \mathcal{L}_\mathcal{G}(\theta_k)\|^2 \right] \leq \|\nabla \mathcal{L}(\theta_k)\|^2 + \frac{\sigma^2}{N_\mathcal{G}}.$$

Taking total expectation and noting $\mathbb{E}[\mathcal{L}_\mathcal{G}(\theta)] = \mathcal{L}(\theta)$, we obtain

$$\mathbb{E}[\mathcal{L}(\theta_{k+1})] \leq \mathbb{E}[\mathcal{L}(\theta_k)] - \tfrac{\eta_k}{2} \mathbb{E}\left[ \|\nabla \mathcal{L}(\theta_k)\|^2 \right] + \tfrac{L}{2} \frac{\eta_k^2 \sigma^2}{N_\mathcal{G}}.$$

Since $\mathcal{L}(\theta)$ is lower bounded (Theorem 1), applying the Robbins–Siegmund theorem (Robbins & Siegmund, 1971) gives

$$\sum_{k=0}^{\infty} \eta_k \mathbb{E}\left[ \|\nabla \mathcal{L}(\theta_k)\|^2 \right] < \infty.$$

Given $\sum_k \eta_k = \infty$, it follows that

$$\liminf_{k \to \infty} \mathbb{E}\left[ \|\nabla \mathcal{L}(\theta_k)\|^2 \right] = 0.$$

If $\eta_k \equiv \eta \leq 1/L$ is fixed, the residual is of order $\mathcal{O}(\eta)$ (from smoothness) and $\mathcal{O}(1/N_{\mathcal{G}})$ (from gradient variance). Hence

$$\limsup_{K \to \infty} \frac{1}{K} \sum_{k=1}^{K} \mathbb{E}\Big[\|\nabla\mathcal{L}(\theta_k)\|^2\Big] \;\lesssim\; \mathcal{O}(\eta) + \mathcal{O}\Big(\tfrac{1}{N_{\mathcal{G}}}\Big).$$

With diminishing step sizes, the algorithm converges to a stationary point of the expected objective; with a small constant step size, it converges to a neighborhood of stationarity whose size depends on $\eta$ and the group size. The advantage momentum does not affect the asymptotic rates, as it only changes the constant $B$ (and hence $L = BL_0$). $\qquad\square$

## C  EXPERIMENT DETAILS

### C.1  REWARD RULES

**Format Reward**   To encourage adherence to structured reasoning, we adopt a binary format reward $R_{format}(o) \in \{0,1\}$, which assigns a reward of 1 if the model response $o$ conforms to the expected template by containing the delimiter sequence "\n </think>\n", and 0 otherwise.

**Cosine Scaled Reward**   We adopt the $R_{cosine\_scaled\_accuracy} \in \{0,1\}$ as expressed in equation 9, which encourages correct outputs with shorter lengths and penalizes incorrect outputs with reduced severity as their length increases, following a cosine annealing schedule in equation 10.

$$R_{cosine\_scaled\_accuracy}(o) = \begin{cases} R_{\text{correct}}(l), & \text{if correct} \\ R_{\text{wrong}}(l), & \text{if wrong} \end{cases}, \tag{9}$$

where

$$\begin{aligned} R_{\text{correct}}(l) &= \alpha_{\min}^c + \frac{1}{2}(\alpha_{\max}^c - \alpha_{\min}^c)\left[1 + \cos\left(\pi\frac{l}{L}\right)\right], \\ R_{\text{wrong}}(l) &= \alpha_{\max}^w + \frac{1}{2}(\alpha_{\min}^w - \alpha_{\max}^w)\left[1 + \cos\left(\pi\frac{l}{L}\right)\right], \end{aligned} \tag{10}$$

**Accuracy Reward**   We adopt a standard accuracy reward $R_{accuracy} \in \{0,1\}$, which assigns a binary reward of 1 for correct response and 0 otherwise, providing a sparse but direct reward signal.

### C.2  TRAINING SETUP

**Training DeepSeek-R1-Distill-Qwen-1.5B**   For training DeepSeek-R1-Distill-Qwen-1.5B on open-rs experiment (Dang & Ngo, 2025), we adopt the $R_{format}$ and $R_{cosine\_scaled\_accuracy}$ reward functions, with respective weights of 1 and 2. We directly perform our AAPO training from the base model without any SFT with the group size of 6 and the per_device_batch_size of 6 with gradient_accumulation_steps of 4 on 2 Nvidia A800 GPUs with 80G VRAM. The system prompt adopted in the RL training is provided below.

> System prompt for training DeepSeek-R1-Distill-Qwen-1.5B base model
>
> You are a helpful AI Assistant, designed to provided well-reasoned anddetailed responses. You FIRST think about the reasoning process as an internal monologue and then provide the user with the answer. The reasoning process MUST BE enclosed within <think>and </think>tags.

**Training Qwen2.5-Math-7B**   For training Qwen2.5-Math-7B (Yang et al., 2024) on sim-plelr_qwen_level3to5 (Zeng et al., 2025), we adopt the $R_{accuracy}$ reward function with a weight of 1. We directly perform our AAPO training without any SFT with the group size of 8 and the per_device_batch_size 8 with gradient_accumulation_steps of 4 on 8 Nvidia A800 GPUs with 80G VRAM. The system prompt adopted in the RL training is provided below.

---

System prompt for training Qwen2.5-Math-7B base model

A conversation between User and Assistant. The user asks a question, and the Assistant solves it. The assistant first thinks about the reasoning process in the mind and then provides the user with the answer, and put your final answer within \\boxed . The reasoning process and answer are enclosed within <think></think >and <answer></answer>tags, respectively, i.e., <think>reasoning process here </think ><answer>answer here </answer>. Note that respond by English, NOT use other languages.

---

**Training Llama series models**    For training Llama-3.2-1B-Instruct and Llama-3.2-3B-Instruct models, we adopt the simplelr_able_level3to5 dataset (Zeng et al., 2025) and the $R_{accuracy}$ reward function with a weight of 1. We directly perform our AAPO training without any SFT with the group size of 8 and the per_device_batch_size of 16 on 4 Nvidia A800 GPUs with 80G VRAM. There is no extra system prompt added when training Llama series models.

## C.3 EVALUATION SETUP

**Evaluation of Qwen series models**    All the reproduced results in Table 1 are evaluated using the lighteval framework (Habib et al., 2023) with vllm backend proposed by Hugging Face. Our evaluation experiments on all benchmarks are conducted on a single Nvidia A800 GPU with 80G VRAM, with system prompt provided in below.

---

System prompt for Qwen series evaluation on all benchmarks

Solve the following math problem efficiently and clearly. The last line of your response should be of the following format: 'Therefore, the final answer is: $\\boxed{{ ANSWER}}$. I hope it is correct' (without quotes) where ANSWER is just the final number or expression that solves the problem. Think step by step before answering.

---

**Evaluation of Llama series models**    All the reproduced results in Table 2 are evaluated using vllm backend for generation and the evaluation script provided by Zeng et al. (2025). Our evaluation experiments on all benchmarks are conducted on 4 Nvidia A800 GPUs with 80G VRAM.

## D    MORE ANALYSIS ON EXPERIMENT RESULTS

We plot the loss curves of AAPO and GPG (Chu et al., 2025) during training DeepSeek-R1-Distill-Qwen-1.5B on open-rs (Dang & Ngo, 2025) in Figure 3. As observed, the loss values for AAPO remain predominantly positive throughout the training process. This indicates that AAPO primarily optimizes the policy by encouraging diverse responses better than the reference, assigning different gradient magnitudes according to the advantage estimated by $\hat{A}_{i,t}^{*}$, thus leading to performance improvements. In contrast, GPG exhibits mostly negative loss values, suggesting that it focuses on suppressing suboptimal responses as its main optimization strategy. It can be clearly seen from the Figure 3 that the training process of AAPO is more stable than GRPO (Shao et al., 2024). These results of the analysis imply that models trained with our proposed AAPO demonstrate stronger generalization, resulting in superior performance as shown in Table 1.

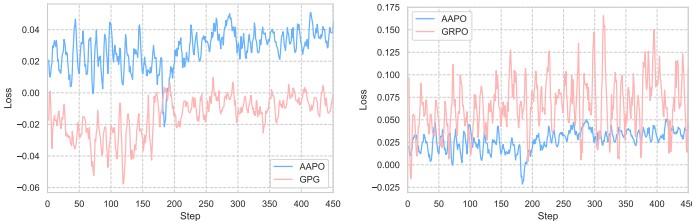

Figure 3: A comparative analysis of training loss between AAPO and GPG, AAPO and GRPO.

# E    EXTRA COMPARISON WITH ORIGINAL RESULTS

Here we present the original results reported in the corresponding papers for Qwen series models.

| Model | AIME24 | MATH-500 | AMC23 | Minerva | OlympiadBench | Avg. |
|---|---|---|---|---|---|---|
| *Qwen 1.5B Models* | | | | | | |
| DeepSeek-R1-Distill-Qwen-1.5B | 28.9 | 83.9 | – | – | – | – |
| GRPO-1.5B (Dang & Ngo, 2025) | 46.7 | 84.4 | 72.5 | 26.8 | 51.3 | 56.3 |
| GPG-1.5B (Chu et al., 2025) | 33.3 | 85.0 | 80.0 | 26.8 | 52.4 | 55.5 |
| Still-3-1.5B-Preview (Chen et al., 2025) | 39.3 | 85.5 | – | – | – | – |
| AAPO-1.5B (Ours) | 33.3 | 86.0 | 80.0 | 30.9 | 53.3 | **56.7** |
| *Qwen 7B Models* | | | | | | |
| Qwen2.5-Math-7B (Yang et al., 2024) | – | 55.4 | – | – | – | – |
| SimpleRL-Zero-7B (Zeng et al., 2025) | 20.0 | 78.2 | 62.5 | 38.6 | 40.4 | 47.9 |
| GPG-7B(Chu et al., 2025) | 33.3 | 80.0 | 65.0 | 34.2 | 42.4 | 51.0 |
| OpenReasoner-Zero-7B (Hu et al., 2025) | – | – | – | – | – | – |
| Eurus-2-7B-PRIME (Cui et al., 2025) | 20.0 | 78.2 | 50.6 | 39.3 | 40.3 | 45.7 |
| Oat-Zero-7B(Liu et al., 2025) | 43.3 | 80.0 | 62.7 | 30.1 | 41.0 | 51.4 |
| AAPO-7B (Ours) | 30.0 | 82.4 | 70.0 | 35.3 | 44.3 | **52.4** |

Table 3: Zero-shot pass@1 performance on mathematical reasoning benchmarks. All reported results in this table are directly adopted from the corresponding papers. Dashes (–) denote unavailable official score.

# F    DISCLOSURE OF LLM USAGE

LLMs are utilized to assist in the drafting, refinement, and wording adjustments of portions of the paper's text. All content was ultimately reviewed and revised by the authors, who are solely responsible for the facts, assertions, and arguments presented herein.

