# OpenReview forum: "AAPO: Enhancing the Reasoning Capabilities of LLMs with Advantage Momentum"
_ICLR.cc/2026/Conference — ICLR 2026 Conference Withdrawn Submission_

### Official Review · Reviewer_gPpc · 2025-10-21

**Soundness:** 4
**Presentation:** 3
**Contribution:** 3
**Rating:** 6
**Confidence:** 4

**Summary:**

This paper introduces Advantage-Augmented Policy Optimization (AAPO), a novel RL algorithm designed to address the limitations of current group relative advantage estimation methods. Specifically, existing GRPO-like advantage estimation can lead to zero gradient or unstable optimization when rewards within a group exhibit low variance or vary significantly, thereby hindering optimization efficiency. AAPO overcomes both issues by augmenting advantage estimation with advantage momentum, which takes a comparison with the reference model. Experiments across multiple benchmarks demonstrate its effectiveness.

**Strengths:**

1. This work analyzes the limitations of group relative advantage estimation in mainstream RL algorithms, clearly motivating the new algorithm.

2. The proposed advantage augmentation algorithm through advantage momentum is simple and effective, overcoming both challenges.

3 . Extensive experiments on challenging math-reasoning benchmarks demonstrate its effectiveness.

**Weaknesses:**

1. While AAPO demonstrates effectiveness in mathematical reasoning, its applicability to other domains or tasks remains underexplored. How does AAPO perform on non-mathematical reasoning tasks?

2. The evaluation would benefit from more consistent and comprehensive comparisons with RL algorithms.

**Questions:**

1. How is the clip threshold in Equation 5 determined, and how does it impact overall performance?

2. A comparative analysis of advantage estimation between the AAPO algorithm and baseline methods is essential. For example, direct evidence should be provided to show that AAPO effectively resolves the issue of zero advantages within groups when responses are uniformly high or low in quality.

3. Figure 2 shows a minimal improvement in optimization stability for AAPO compared to the baseline, along with a slower convergence rate.

---

### Official Review · Reviewer_qxLT · 2025-10-27

**Soundness:** 2
**Presentation:** 2
**Contribution:** 2
**Rating:** 2
**Confidence:** 4

**Summary:**

This paper proposes a novel reinforcement learning algorithm called Advantage-Augmented Policy Optimization (AAPO), which aims to enhance the reasoning capabilities of large language models (LLMs). By introducing a momentum-based advantage estimation derived from a reference model, a correction term is incorporated into the advantage calculation, thereby preserving effective gradients and preventing training stagnation.

**Strengths:**

AAPO is a RL algorithm that optimizes the cross-entropy (CE) loss using advantages enhanced through a momentum-based estimation scheme. This approach is simple and this paper is easy to follow. Experimental results on multiple mathematical reasoning benchmarks and model series demonstrate the superior performance of AAPO.

**Weaknesses:**

1. Novelty: The issue of vanishing gradients in GRPO training is a classic and extensively studied topic, as seen in works such as GRPO-LEAD[1], DRA-GRPO[2], VL-Rethinker[3], and Dr.GRPO. These studies all identified the vanishing gradient problem in GRPO's advantages and proposed corresponding improvements, with their publication dates preceding the ICLR deadline by over five months. Since the authors' core innovation lies solely in modifying the advantages, they need to clarify the key distinctions and advantages of their work compared to these related studies to strengthen its novelty. Additionally, the baselines used for comparison are outdated and should be replaced with more recent ones.
2. Completeness: The authors only conducted experiments on five mathematical benchmarks, without validating generalization. The ablation studies are simplistic and lack case analyses of AAPO’s reasoning behavior or other meaningful analyses beyond performance metrics (e.g., gradient-related analyses to substantiate the mitigation of vanishing gradients). Furthermore, the effectiveness of AAPO on more advanced methods beyond PGP or GRPO (e.g., DAPO, GSPO) has not been analyzed.
3. Theoretical Analysis: The theoretical analysis lacks insightful content. The discussion in Section 5 covers widely known  concepts and appears to serve as filler material.
4. Presentation: The writing is unclear and lacks essential hyperparameters. In Table 1, "DeepSeek-R1-Distill-Qwen-1.5B" is presented as a baseline, while Figure 2 introduces the "Training process of DeepSeek-R1-Distill-Qwen-1.5B," creating ambiguity about which base model was used for training in Figure 2.

**Questions:**

1. The method proposed in the paper does not intuitively address the vanishing gradient issue: For challenging problems, both r_ref and r_theta are zero. Does this approach hold any significance in such cases?
2. The authors did not specify the values of the rewards and treated them as discrete values. Since r_θ – r_ref is also a common discrete value, the clip operation seems intuitively unnecessary and could simply be replaced by multiplying a coefficient. The parameters used for clipping are also not explained in the paper.
3. In the bottom-left graph of Figure 2, why does the GRPO format reward not approach 1 even after 400 training steps? This is unreasonable in normal training and may be due to improperly set GRPO parameters or an excessively low max_response_length.

---

### Official Review · Reviewer_tDff · 2025-10-31

**Soundness:** 2
**Presentation:** 3
**Contribution:** 2
**Rating:** 4
**Confidence:** 4

**Summary:**

The paper proposes AAPO, a reinforcement-learning algorithm for LLM post-training that tackles a failure mode of group-relative advantage estimation (as in GRPO/GPG) where advantages collapse toward zero when within-group reward variance is low, yielding no gradients. AAPO keeps the cross-entropy loss form but multiplies token losses by an augmented advantage: the normalized group-relative advantage plus a clipped "advantage momentum" term defined as the reward difference between a response sampled from the policy and a response sampled from a frozen reference model. This preserves a non-zero RL signal late in training. The authors reports gains over GRPO/GPG on several math-reasoning benchmarks (AIME24, MATH-500, AMC23, Minerva, OlympiadBench) across multiple model families (Llama-1B/3B, DeepSeek-R1-Distill-Qwen-1.5B, Qwen2.5-Math-7B).

**Strengths:**

1. The paper presents a clear diagnosis of the optimization issue in GRPO (and GPG). The formalization is well-written and also motivates the fix.
2. The proposed method is a simple and implementable modification to existing algorithm. It augments CE with a constant per-token weight that adds a clipped reward ratio, which can be directly integrated with any of the existing GRPO-flavored algorithm.
3. The experiments (and ablations) show effective momentum clipping, preserving high group variance. the training curves seem to also suggest smoother optimization.

**Weaknesses:**

1. The proposed method adds significant inference overhead via two groups of response sampling (from both the policy and reference). An analysis of the benefit under compute overhead is missing. I do recognize that the paper emphasizes training efficiency, but training efficiency cannot be decoupled from the actual compute spent.
2. The method introduces additional hyperparameters such as delta_low and delta_high, which seem to be fixed in the experimental setup. It's unclear how the algorithm itself is sensitive to these hyperparameters and how to set them in practice.
3. I find the term "advantage momentum" to be somewhat misleading, which I interpreted differently in my first pass of the paper. This term seems to be a reward gap w.r.t. a frozen reference and not a temporal smoothing mechanism toward the reward / gradients, right? The authors can correct me if I am wrong about this.
4. All experiments seem to be based solely on math tasks. No evidence of domain generalization is provided.

**Questions:**

1. What is the effect of group size G, policy/reference sampling temperatures, and decoding strategies on the momentum term and stability? And how is the method compared with purely increasing group size to 2x?
2. How sensitive is AAPO to the reference model (e.g., initial checkpoint vs. intermediate ones)?
3. In Table 1 are the GRPO and GPG using the same training prompts?

---

### Official Review · Reviewer_fGA3 · 2025-11-01

**Soundness:** 2
**Presentation:** 3
**Contribution:** 2
**Rating:** 2
**Confidence:** 5

**Summary:**

This paper uses the reward difference between the policy and a frozen reference model to augment group-relative advantage estimates and to directly optimize a weighted cross-entropy loss. This approach enables continual model updates: even if every response in a sampled group is correct, the training signal remains non-vanishing and the model updates stably. Empirically, the method delivers consistent improvements across diverse base models and benchmarks.

**Strengths:**

1 AAPO provides an apparently reasonable stability and convergence bound as theoretical support.

2. The overall framework of this paper is clear.

3 Experiment results look good.

**Weaknesses:**

## For related work
Among GRPO-based methods, many explicitly tackle the vanishing-advantage problem—for example, DAPO [1] and VL-ReThinker [2]—but the paper’s Introduction and Related Work sections do not discuss this literature, which weakens the motivation for the proposed approach.

## For method

In GRPO, when the advantages $\approx$ 0, it typically means the sampled response is either almost entirely correct or almost entirely wrong. Under the paper’s setup, even in the “all-correct” case the update still relies on outcome-based advantage momentum. However, prior work [3] has reported that during RL fine-tuning for some cases, response length can keep increasing. If length continues to grow under the “all-correct” condition, the model may obtain reward from redundant information. Whether such length-inflation gains should still be regarded as optimization “in the right direction” is a question.

## For theory
Section 4 is written in an extremely confusing and unprofessional way. I suggest the authors look at how other papers structure a theorem. In addition, the discussion of the two theorems is minimal—I don’t see what the authors intend to emphasize. I recommend adding remarks to further explain and discuss.

Although the two bounds are standard in form and their conditions are indeed satisfied, Theorem 1 appears overly loose. I do not think this upper bound is sufficient to demonstrate the stability of AAPO.

## For experiment

The paper indeed leaves too many details unspecified. For example: how was this result measured? Over how many runs was the average computed? Did baselines use the same data as method? I hope the authors can provide more details.


[1] Yu, Q., Zhang, Z., Zhu, R., Yuan, Y., Zuo, X., Yue, Y., ... & Wang, M. (2025). Dapo: An open-source llm reinforcement learning system at scale. arXiv preprint arXiv:2503.14476.


[2] Wang, H., Qu, C., Huang, Z., Chu, W., Lin, F., & Chen, W. (2025). Vl-rethinker: Incentivizing self-reflection of vision-language models with reinforcement learning. arXiv preprint arXiv:2504.08837.


[3] Liu, Z., Chen, C., Li, W., Qi, P., Pang, T., Du, C., ... & Lin, M. (2025). Understanding r1-zero-like training: A critical perspective. arXiv preprint arXiv:2503.20783.

**Questions:**

## For proof

The authors should list all assumptions for the theoretical results up front in a dedicated section, rather than placing them under the first proof. The second proof also relies on several assumptions. Presenting the assumptions together at the beginning will make the paper more professional.

Other can be seen in weakness.

---

### Official Review · Reviewer_irju · 2025-11-04

**Soundness:** 2
**Presentation:** 2
**Contribution:** 2
**Rating:** 4
**Confidence:** 3

**Summary:**

The paper addresses a well‑known issue in group‑relative advantage training (e.g., GRPO/GPG) for RL post‑training of LLMs: when rewards within a sampled group become similar, estimated advantages collapse toward zero and gradients vanish; when they vary widely, advantages can be high‑variance and unstable. The authors propose Advantage‑Augmented Policy Optimization (AAPO), which keeps a group‑relative term but adds a clipped “advantage momentum”—the reward difference between the current policy and a frozen reference model for the same prompt. The combined weight then scales a token‑level cross‑entropy objective. The paper includes stability and convergence arguments under standard boundedness assumptions and reports consistent average Pass@1 gains over GRPO/GPG across multiple math‑reasoning benchmarks (AIME24, MATH‑500, AMC23, Minerva, OlympiadBench) and model sizes (Llama‑1B/3B, Qwen‑1.5B/7B), with ablations indicating that clipping the momentum term improves stability and final accuracy.

Disclosure: I used assistive writing tools to draft this review; all evaluations and judgments are my own.

**Strengths:**

Clear motivation and diagnosis. The paper neatly explains why group‑relative estimators tend to produce near‑zero gradients late in training (all‑good/all‑bad groups) and unstable updates when group variance is high.

Simple, drop‑in mechanism. Advantage momentum (policy‑vs‑reference reward delta) is easy to implement, slots into GRPO/GPG‑style pipelines, and does not introduce a critic.

Theoretical sanity checks. Stability (bounded step sizes / lower‑bounded objective) and convergence are spelled out, which helps justify the augmentation.

Breadth of empirical evidence on math. Multiple model scales and several standard math suites show consistent average gains; training curves suggest smoother dynamics and reduced loss variance vs GRPO.

Useful ablations. Removing the clip on the momentum term degrades stability/accuracy, clarifying that clipping is functionally important.

**Weaknesses:**

Assumption realism in the theory.
Stability relies on a strictly positive lower bound for the group standard deviation. The paper does not diagnose when group std approaches zero in practice (ironically, the problematic regime). Training‑time diagnostics would help connect the theory to observed dynamics.

Narrow domain coverage.
Results focus on math reasoning with rule‑based verifiable rewards. Claims of general utility would be stronger with at least a small pilot on non‑math tasks (e.g., code generation or other verifiable domains). Or autoformalization or theorem proving in lean. The latter would impress me because of it's novelty eg really open compared to general coding.

Statistifal significance. no confidence values or p-values reported.

**Questions:**

Non‑math pilot. Add a small code‑generation or other verifiable task to probe generality. Especially in autoformalization or theorem proving.

Statistifal significance. The results improvements are unconvincing and no confidence intervals or p-values are reported. I'd like to know if this results are truly significant. You can use eg the size of the benchmark eval batch size to estimate SE --> CI.

I'd like to see a better motivation for the algorithm, momentum seems not principled and the improvements are very modest. Why didn't you instead try an ADAM style improvement then? or a muon or shampoo etc. Those ablations are just as random (to me) and comparing them would provide convincing empirical depth to your paper. Prompt gpt5-pro with this suggestion and try it. If it works I'd increase my score. If it doesn't and it's interesting then maybe I would too, at least it would motivate momentum better and make it seem less random.

---

### Note · Authors · 2025-11-12

I have read and agree with the venue's withdrawal policy on behalf of myself and my co-authors.